# Application of Proximal Optical Sensors to Fine-Tune Nitrogen Fertilization: Opportunities for Woody Ornamentals

**Jolien Bracke** [1,2,3], **Annemie Elsen** [3], **Sandy Adriaenssens** [4], **Lore Schoeters** [4], **Hilde Vandendriessche** [2,3] **and Marie-Christine Van Labeke** [1,*]

1 Department Plants and Crops, Faculty of Bioscience Engineering, Ghent University, Coupure Links 653, 9000 Ghent, Belgium
2 Department of Biosystems, Faculty of Bioscience Engineering, KU Leuven, Kasteelpark Arenberg 20, 3001 Leuven, Belgium
3 Soil Service of Belgium, Willem de Croylaan 48, 3001 Leuven, Belgium
4 PCS Ornamental Plant Research, Schaessestraat 18, 9070 Destelbergen, Belgium
* Correspondence: mariechristine.vanlabeke@ugent.be; Tel.: +32-9-264-60-71

**Abstract:** Today, high amounts of residual nitrogen are regularly being reported in the open field production of hardy nursery stock. In some cases, excessive fertilizers or side-dressings are applied when circumstances are not favorable for uptake. Aquatic as well as terrestrial ecosystems are sensitive to enrichment with nutrients, but growers also benefit when losses are avoided. In this study, the potential of proximal optical sensors to optimize nitrogen fertilization was investigated in four woody species: *Acer pseudoplatanus* L., *Ligustrum ovalifolium* Hassk., *Prunus laurocerasus* 'Rotundifolia' L. and *Tilia cordata* Mill. For three consecutive growing seasons, plants were grown under three different fertilization levels to generate different nitrogen contents. Plant growth and nitrogen uptake were monitored regularly and combined with sensor measurements including Soil Plant Analysis Development (SPAD), Dualex and GreenSeeker. Here, we show that optical sensors at the leaf level have good potential for assisting growers in the sustainable management of their nursery fields, especially if leaf mass per area is included. Nevertheless, care should be taken when plants with different leaf characteristics (e.g., wax-layer, color, and leaf thickness) are measured. When all measuring years were considered, high correlations ($R^2 \geq 0.80$) were found between area-based foliar nitrogen content and its non-destructive proxy (i.e., chlorophyll)measured by Dualex or SPAD. Based on our results, we recommend a relative rather than absolute approach at the nursery level, as the number of species and cultivars produced is very diverse. Hence, knowledge of absolute threshold values is scarce. In this relative approach, a saturation index was calculated based on the sensor measurements of plants grown in a reference plot with an ample nitrogen supply.

**Keywords:** nursery management; SPAD; Dualex; GreenSeeker; *Acer pseudoplatanus*; *Ligustrum ovalifolium*; *Prunus laurocerasus*; *Tilia cordata*

## 1. Introduction

In recent years, more and more attention has been paid to the impact of arable crops on the environment. The shift from the highest possible production, regardless of potential environmental consequences, to a sustainable and crop-based fertilization strategy is the focus of much scientific research today. Since 1991 in Europe, the Nitrate Directive has imposed requirements on its member states concerning water quality by the prevention of nitrates from fertilization which pollute ground and surface water [1]. In Flanders, this led to fertilizer restrictions and the implementation of a legal maximum mineral N ha$^{-1}$ in the upper 90 cm soil between October 1 and November 15, depending on

crop and soil type. Farmers are encouraged to apply well-considered nitrogen fertilization. However, high amounts of residual nitrogen are regularly being reported for the open field production of ornamental trees and shrubs. Opposed reasons are an incorrect nitrogen fertilization strategy and a mismatch between timing of N fertilization and N uptake by plants [2].

Despite diverse research on fertilizer rates in woody ornamentals, there is a lot of variation in recommended fertilizer use. Some authors report little or no effect of fertilizers on ornamental plant quality (e.g., height and length) [3,4], others do find effects [5,6], and others report mixed results depending on the species [7]. According to Rose, suggested annual N rates vary between 48 and 287 kg N ha$^{-1}$ [8]. The author emphasized that these rates are based on research from the 1950s to the 1970s when maximizing the fertilizer response was the primary goal and other parameters such as soil N content, yield potential and field history were not taken into account. This is indeed very important for open field production with big differences in soil organic matter content, soil type, water availability, and thus natural N mineralization. Moreover, hardy nursery growers often grow a wide range of species and varieties differing in age and planting distance on one ore multiple fields [9]. Though some reports suggest low, medium, and high rates within the range for broadleaf evergreens, conifers, and deciduous plants, respectively, more research is still needed. Studies in the past focused on the N uptake patterns of woody plants with episodic rather than continuous growth, characterized by alternate episodes of shoot and root growth. Gilliam and Wright found that fertilizer applied during inactive shoot growth resulted in a greater N content in *Ilex crenata* [10]. In addition, protein concentrations in *Ligustrum japonicum* decreased during shoot growth but increased in the roots during root growth; additionally, labelled $^{15}$N denoted the translocation of N from roots to shoots during shoot growth [11]. Similar results were reported by Hershey and Paul for *Euonymus japonica* [12]. The timing of N fertilization is essential to assure the supplied N is used by the plants; however, changes in growth rate over time at constant climatic conditions makes it even more difficult to decide when an additional side dressing is appropriate.

Since non-controllable parameters (mainly weather conditions) can severely influence nitrogen availability and uptake, it is highly questionable if the knowledge of total N uptake is sufficient for fertilizer advice. An outcome can provide guidelines for growers, but the need for side dressing is always dependent on the actual N statuses of both soil and crops. Foliar analysis is a well-established method to assist the diagnosis of N deficiency. Since N is incorporated in chlorophyll, foliar N content is positively correlated with chlorophyll content [13,14]. Because non-destructive sensing with chlorophyll meters (e.g., Soil Plant Analysis Development, SPAD) instead of a destructive, expensive and time-consuming lab-analysis is more convenient, foliar N analysis is being replaced by sensor measurements in arable crops. However, these crops are typically uniform and straightforward compared with woody ornamentals. Here, nutrient concentration depends on crown position, plant and leaf age, competition/light, and even provenance [15]. In addition, many authors have reported that correlations between sensor measurements and actual nutrient content can be interfered by, e.g., cultivar, leaf age, and other characteristics such as dry leaf mass per area (LMA) [16–20].

Time of the year is an important factor as well. The most appropriate moment for detecting N deficits would be during periods of maximum growth when reserves are depleted and demands are large [15]. When trees or shrubs become dormant, mobile nutrients such as N are translocated to perennial storage organs, and foliar N concentration is less indicative for the N status of the plant. The diagnosis of N deficiency by the means of foliar analysis can be effective, but the determination of the optimum foliar nutrient concentration for all different species grown in a nursery is expensive, time-consuming, or even impossible when cultivars are considered. Since the main interest is not the actual chlorophyll or nitrogen content but whether there is a deficiency, a possible solution to the absence of species/cultivar specific thresholds is the use of a relative assessment. Therefore, instead of using absolute sensor values, sensor measurements are being normalized by dividing them into those of a non-limiting reference area receiving ample N fertilization [21,22]. In this way, since the reference area will struggle similarly—as shown by Blackmer et al. for corn [23] and by Westerveld et al. for

carrot, onion, and cabbage [24]—other affecting stress factors such as water stress, pests, or diseases are eliminated. The result is a saturation index (SI) that can be used to support N management decisions. The results of Varvel et al. suggest an action threshold of 90% to achieve maximum yield in corn [25].

More recently, the estimation of leaf epidermal polyphenols content (EPhen) has been used as an indicator for N stress, since the synthesis of these compounds is upregulated when stress, e.g., N shortage, occurs. EPhen can be estimated non-destructively and simultaneously with chlorophyll by a Dualex sensor [26]. The ratio of both estimations is called the nitrogen balance index (NBI) and has been found more useful compared with each estimation separately [27]. Another possibility which allows the user to integrate a larger area and thus make a more representative assessment of the crop is the use of canopy reflectance sensors, such as the GreenSeeker (Trimble Inc., Sunnyvale, CA, USA) which calculates the normalized difference vegetation index (NDVI, (IR−R)/(IR + R)). The NDVI is generally related with the amount of green biomass [28]. This approach makes it possible to evaluate entire fields and can support management decisions on a small scale. A disadvantage, however, is the fact that background/soil scattering is present, especially for plants with a large planting distance, such as the production of avenue trees. Furthermore, the NDVI tends to saturate when the canopy closes [29].

The SPAD meter has been researched on some woody plants such as the sycamore (*Acer pseudoplatanus*), English oak (*Quercus robur*), and European beech (*Fagus sylvatica*) [30]. Good correlations between N and SPAD were reported, but these were species specific and only measured at one point in time. SPAD and Dualex were used by Meyer et al. on sycamore, European beech, birch (*Betula pendula*) and ash (*Fraxinus excelsior*) trees, as well as on *Lagerstroemia indica*, *Callicarpa bodinieri*, *Robinia pseudoacacia* and *Viburnum tinus* [31]. The authors found that chlorophyll readings increased moderately with an increasing LMA, which increased with irradiance because of the variability in the crown position for every species. Furthermore, a good correlation ($R^2$ = 0.78) between mass-based estimated chlorophyll (= chlorophyll/LMA) and foliar N content was reported for four species over a timespan of two months (*Lagerstroemia*, *Callicarpa*, *Robinia* and *Viburnum*). Demotes-Mainard et al. used both sensors for three ornamental shrubs: *Lagerstroemia indica*, *Callicarpa bodinieri* and *Viburnum tinus* [32]. Foliar N content was found to correlate best ($R^2$ resp. 0.85, 0.53 and 0.83) with EPhen values, but it was species-dependent. The NBI improved the relation between foliar N content and SPAD measurement, but EPhen alone resulted in higher correlation coefficients. Variations in LMA values negatively affected the relationship between SPAD and N.

The objectives of this study were (1) to determine if non-destructive sensor measurements can predict N content and thus indicate a N deficiency for four commonly grown woody plant species (*Acer pseudoplatanus*, *Ligustrum ovalifolium*, *Prunus laurocerasus* 'Rotundifolia' and *Tilia cordata*); (2) to determine if correlations are species and/or growing season dependent; and (3) if including LMA or the saturation index approach could increase applicability. To do so, we analyzed N content on a regular basis during three entire growing seasons on plots with different fertilization levels. The validation of these objectives is an essential step to determine the utility of non-destructive proximal sensors as an on-farm decision support system in commonly grown woody species.

## 2. Materials and Methods

### 2.1. Sampling Site, Planting Material and Experimental Design

Experimental work was conducted at PCS Ornamental Plant Research, Destelbergen, Belgium (51°04′18.3″N, 3°49′01.5″E). The soil had a fine sand texture and an organic carbon content of 1.6%. Three experiments were carried out for three consecutive years (2016–2018). The time scale is represented in Days of the Year (DOY), e.g., 1 for the 1 January and 365 for the 31 December for non-leap years Julian Day (JD) (see Supplementary Table S1 for date conversion).

One-year-old seedlings of *Acer pseudoplatanus* L. (deciduous, Aceraceae) and rooted cuttings of *Prunus laurocerasus* 'Rotundifolia' L. (evergreen, Rosaceae) were planted for three consecutive years on 31 March 2016, 3 April 2017, and 13 November 2017, respectively, and monitored for one growing season. An autumn planting was chosen for the third growing season to stimulate early rooting and thus reduce drought stress potential. The *P. laurocerasus* cuttings were rooted in 2016 and had already grown for one year in 2017 and 2018. One-year old cuttings of *Ligustrum ovalifolium* Hassk. (semi-evergreen, Oleaceae), were planted on 13 March 2016 and 3 April 2017. In autumn 2017, the now two-year-old *L. ovalifolium* plants were cut back strongly for a new growing season. Harvested plants were replaced with other two-year-old plants from a commercial nursery and cut back to the same size as the others. These plants were labeled to avoid using them as sampling plants in 2018. *Tilia cordata* Mill. (deciduous, Tiliaceae) was planted for avenue tree production on 13 March 2016 and monitored for three growing seasons. Shoots were pruned yearly to support crown development. In 2017, two extra lines of *T. cordata* of similar age were planted next to this original plot as border plants.

For each species, a randomized 1-factorial design with three replicates for each N level was used. Specific planting densities and plot dimensions are given in Supplementary Figure S1. All experimental plants were surrounded by border plants.

The top soil (0–30 cm) before planting at the end of March contained 18 kg ha$^{-1}$ of mineral N in 2016. In 2017, three different soil samples were taken and averaged for *T. cordata* and *P. laurocerasus* separately. The top soil of each species contained 5 kg N of ha$^{-1}$. For the newly established experimental field, where *A. pseudoplatanus* and *L. ovalifolium* were planted in 2017 (see Supplementary Figure S1), N levels were under the detection limit (< 2 kg N ha$^{-1}$). In 2018, three soil samples were taken per species. On average, the top soil contained, respectively 3, 13, 12 and 12 kg ha$^{-1}$ for *A. pseudoplatanus*, *L. ovalifolium*, *P. laurocerasus* and *T. cordata* in 2018.

Nitrogen soil dressing treatments were applied to provide plants with N levels ranging from deficient to excessive for the three consecutive years (Table 1). The N rate in 2017 and 2018 was adapted based on results of 2016 or 2017 and soil analysis prior to starting. Calcium ammonium nitrate (27% N, Scoriethom, Belgium) was applied in 2016, and *Tropicote*® (15.5% N, Yara, The Netherlands) was applied in 2017 and 2018. Different applications (3–4 per year) occurred with intervals of 3–5 weeks, depending on weather forecasts (exact dates in Supplementary Table S2). Before planting in 2016, each plot was fertilized with 5.4 kg of Patentkali (30% $K_2O$, 10% MgO and 42% $SO_3$, Scoriethom, Belgium). In 2017, an additional dose Patentkali of 1.4 kg per plot (*T. cordata* and *P. laurocerasus*) and 1.22 kg per plot (*A. pseudoplatanus* and *L. ovalifolium*) was given in May. All fertilizers were applied manually and in granular form.

**Table 1.** Overview of applied N fertilizer rates (kg N ha$^{-1}$) applied as soil dressing. Number of applications in brackets.

| | 2016 | | | 2017 | | | 2018 | | |
| N Treatment | N0 | N1 | N2 | N0 | N1 | N2 | N0 | N1 | N2 |
| --- | --- | --- | --- | --- | --- | --- | --- | --- | --- |
| Species/Fertilizer | CAN (3) | | | Tropicote® (4) | | | Tropicote® (3) | | |
| *A. pseudoplatanus* | 0 | 89 | 179 | 0 | 130 | 260 | 0 | 100 | 150 |
| *L. ovalifolium* | 0 | 89 | 179 | 0 | 33 | 67 | 0 | 75 | 113 |
| *P. laurocerasus* | 0 | 89 | 179 | 0 | 33 | 67 | 0 | 75 | 113 |
| *T. cordata* | 0 | 89 | 179 | 0 | 33 | 67 | 0 | 33 | 50 |

Abbreviations: CAN, calcium ammonium nitrate. N0: Deficiency treatment; N1: Standard treatment; N2: High N rate treatment.

The three growing seasons were characterized by contrasting weather conditions (Supplementary Table S3). Spring was very wet in 2016, but summer and autumn were rather dry. September 2016 was warm. Spring 2017 was dry, and 2018 was defined by a very dry summer and autumn and hot summer temperatures. A sprinkler system was installed in 2017 and 2018, but it turned out to be insufficient to avoid water stress during the driest and hottest months during the summer of 2017 and 2018. In 2017,

all plants received ± 6 mm every two weeks in May and June. In 2018, 9 mm was given the end of June, two times of 16 mm (July 18 and 30), and 29 mm on August 7. Weed and pest management were done according to standard horticultural practices.

## 2.2. Plant Measurements

The plant lengths of 5 randomly chosen labeled *A. pseudoplatanus* and *T. cordata* trees were measured every 3 weeks, and the shoot lengths of *L. ovalifolium* and *P. laurocerasus* plants (5 plants, 4 shoots each plant) were measured every 2 weeks in 2016 and 2017. In 2018, the plant and shoot length of all species was measured tri-weekly.

Aboveground biomass increase was measured by periodic biomass samplings by removing representative plants in each replicate plot (exact numbers in Table 2). *A. pseudoplatanus* sampling occurred at the same frequency as growth measurements, but the other species were only harvested twice in 2016. In 2017 and 2018, all species were sampled regularly except *T. cordata* samplings, where sampling was restricted to twice a year. Dry mass (DM, g) was determined after oven drying at 60 °C for 48 h or until a constant weight was achieved. Afterwards, samples were ground, and the total plant N content of ground subsamples was determined by Dumas analysis using an elemental analyzer (Vario MAX CNS, Elementar, Germany). In this procedure, $N_2$ and $CO_2$ are released by pyrolysis and subsequent combustion. $CO_2$ is absorbed over KOH columns, and the residual volume of $N_2$ is measured afterwards [33].

**Table 2.** Numbers of plants sampled per plot (= replicate) at each sampling moment. There were three replicates per fertilization level.

| Year | *A. pseudoplatanus* | *L. ovalifolium* | *P. laurocerasus* | *T. cordata* |
|------|---------------------|------------------|-------------------|--------------|
| 2016 | 1 | 1 | 1 | 1 (2)[1] |
| 2017 | 3 | 2 | 1 (2)[1] | 1 |
| 2018 | 5 (10)[1] | 2 (5)[1] | 2 (5)[1] | 1 (2)[1] |

[1] Numbers in brackets indicate larger numbers of plants sampled per plot at the last sampling moment of that growing season.

## 2.3. Optical Measurements at Leaf Level

In 2016 and 2017, *A. pseudoplatanus* and *T. cordata* leaves were measured every 3 weeks, and *L. ovalifolium* and *P. laurocerasus* leaves were measured every 2 weeks. In 2018, all sensor measurements occurred tri-weekly. At each sampling moment, 10 leaves per plot (= 30 leaves per N treatment) per cultivar were selected in 2016 and 2017, and 15 (= 45 leaves per N treatment) were selected in 2018. Only the youngest, fully expanded, and sun-exposed leaves were measured (Figure 1), and maximum one leaf per plant was sampled.

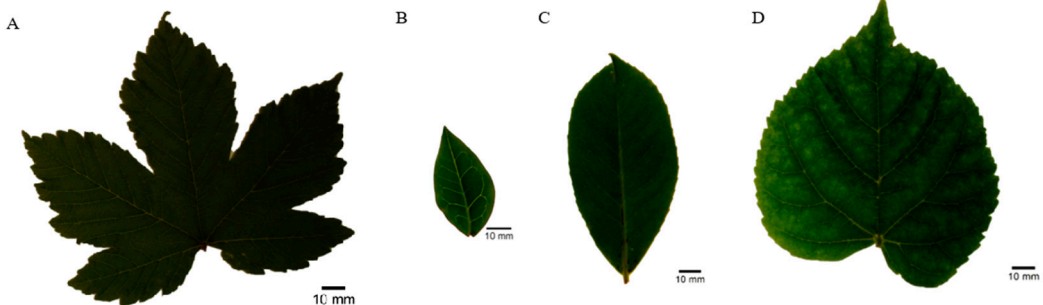

**Figure 1.** Adaxial sides of the leaves of *Acer pseudoplatanus* (**A**), *Ligustrum ovalifolium* (**B**), *Prunus laurocerasus* 'Rotundifolia' (**C**) and *Tilia cordata* (**D**).

A SPAD-502 chlorophyll meter (Konica Minolta, Osaka, Japan) was used in 2016 and 2017 to estimate area-based chlorophyll content. Per leaf, three different readings were taken at the adaxial side of the leaf, avoiding the mid-rib. During the second year, a Dualex Scientific meter (FORCE-A, Orsay, France) was additionally used to estimate area-based chlorophyll content (Chl) and epidermal polyphenols (EPhen). Two measurements were taken on both the adaxial and abaxial sides of the same leaves used for SPAD, and these measurements were averaged to estimate Chl. For the total EPhen, adaxial and abaxial EPhen was summed. The nitrogen balance index (NBI) and a saturation index (SI) were calculated using the following equations:

$$NBI = Chl/EPhen, \tag{1}$$

$$SI\ N0 = sensor\ output\ N0\ /\ sensor\ output\ N2, \tag{2}$$

$$SI\ N1 = sensor\ output\ N1\ /\ sensor\ output\ N2, \tag{3}$$

Leaf dry mass was determined by weighing the leaves after drying at 60 °C for 48 h. Digital leaf images were analyzed by ImageJ software (version 1.51j8) to obtain leaf area in 2017–2018. Leaf mass per area (LMA, g m$^{-2}$), was calculated as the ratio of dry mass to leaf area. For foliar N analysis, the dried leaves were pooled per plot, ground, and stored until analysis. The mass based N content ($N_M$) of the samples was determined by dry Dumas combustion analysis using an elemental analyzer (Vario MAX CNS, Elementar, Germany). By multiplying $N_M$ with the LMA, area-based N content ($N_A$) was obtained.

### 2.4. Optical Measurements at Canopy Level

All plant rows from each plot were scanned at the same frequency as measurements at the leaf level with a GreenSeeker to obtain the NDVI (GreenSeeker RT100, NTech Industries, Ukia, CA, USA). The sensor-canopy distance was kept between 80 and 100 cm (90 ± 10 cm) to enable stable sensor output [34]. The sensor was mounted on a telescoping handheld pole and held horizontally above the canopy while measuring. Measurements were taken within 2 h from local solar noon to minimize influences by solar irradiation differences [35]. For *T. cordata*, due to stem length increases, measurements with the GreenSeeker were only possible in 2016. The SI based on the NDVI was calculated similarly as described in Equations (2) and (3).

### 2.5. Statistical Analysis

Statistical analysis was carried out with R version 3.4.1 software [36]. Data are presented as means ± standard errors. The effect of N treatments on the studied plant variables was assessed using an analysis of variance (ANOVA) followed by Tukey's honest significant difference (HSD) post hoc test ($p \leq 0.05$). The normal distribution and the homoscedasticity of variance assumptions were checked using Shapiro–Wilk and Bartlett tests, respectively. In some cases, when normality or homoscedasticity was severely violated, the data were ln-transformed. Pearson's correlation coefficients (*r*) and regression techniques were performed to analyze the correlation between optical sensor measurements and N content in leaves and plants, using all the sampling dates and replicates for the cultivars considered. Analyses of covariance (ANCOVA) were conducted to test cultivar dependency on the relation between sensor measurements and destructively measured parameters.

## 3. Results

### 3.1. Effects of Fertilization Levels on PlantGgrowth, Quality (Height and Biomass) and N Uptake.

Table 3 displays an overview of all final plant characteristics for all N treatments and growing seasons. Though the N0 (deficiency treatment) treatment resulted in the smallest final biomass for all species and years, little significant differences were detected. In-season biomass accumulation and

length increase can be found in the Supplementary File (Figures S2 and S3). *A. pseudoplatanus*'s biomass accumulation was higher in 2016 compared with 2017 and 2018. Though length growth slowed down around DOY 210, biomass kept increasing. 2017 and 2018, characterized by drier weather conditions compared to 2016, brought forth plants with similar characteristics (height and biomass). In contrast with 2018, the fitted curve of 2017 increased slowly but showed a sharp rise around DOY 210, while the biomass in 2018 was then already at its maximum. The higher slope in the beginning of 2018, due to planting in autumn of 2017, nevertheless resulted in similar plants. *A. pseudoplatanus*'s absolute growth was less vigorous for N0 at most measuring moments, resulting in the smallest final plant length each season (Table 3). This was particularly clear at JD 196 in 2016. In 2018, growth of *A. pseudoplatanus* fell back around DOY 200. *L. ovalifolium* shoots were clearly longer and higher in biomass in 2018 (Table 3), as these plants were in their second growing year. Though they were cut back, their well-developed rooting system resulted in a good regrowth. *L. ovalifolium* showed an episodic growth pattern with two growth flushes in 2016: One distinct peak at JD 225 and one smaller peak around DOY 250–264. In 2017, the episodic character was less profound. In 2018, it was not visible; absolute growth was around 20 cm in the first three weeks and decreased gradually during the growing season. *P. laurocerasus*' plant material in 2016 consisted of rooted cuttings compared to one-year-old and thus more lignified plants in 2017 and 2018, which explains the difference in biomass. Individual shoot length, however, was largest in 2016 (Table 3). *P. laurocerasus*' episodic character clearly manifested in growth flushes in all three growing seasons. Though only significant for *A. pseudoplatanus* in 2018 and *P. laurocerasus* in 2017, N0 tended to result in the lowest biomass accumulation and total aboveground N uptake for all species except *T. cordata*. The highest tree length after three growing seasons of *T. cordata* was found for N1 (standard treatment), and the smallest was found for N0 (Table 3). In the last two years, growth was maximal around DOY 150.

**Table 3.** Final plant quality measurements and N uptake of *A. pseudoplatanus*, *L. ovalifolium*, *P. laurocerasus* and *T. cordata*. Values are means ± standard errors. Values labelled by different letters significantly differ at $p \leq 0.05$ (Tukey's honest significant difference (HSD)-test).

| Species | Year | DOY | N Treatment | Applied N (kg ha$^{-1}$) | Height (cm) | Aboveground Biomass (g plant$^{-1}$) | Aboveground N Uptake (kg ha$^{-1}$) |
|---|---|---|---|---|---|---|---|
| *A. pseudoplatanus* | 2016 | 277 | N0 | 0 | 72.0 ± 3.8b | 26.5 ± 4.4a | 131.2 ± 35.9a |
| | | | N1 | 89 | 98.5 ± 6.9a | 48.9 ± 7.7a | 208.9 ± 37.5a |
| | | | N2 | 179 | 88.9 ± 3.2ab | 42.6 ± 5.9a | 198.5 ± 35.1a |
| | 2017 | 278 | N0 | 0 | 46.3 ± 2.4a | 17.2 ± 1.7a | 75.4 ± 10.0a |
| | | | N1 | 130 | 52.2 ± 6.4a | 20.4 ± 3.6a | 102 ± 21.7a |
| | | | N2 | 260 | 58.2 ± 9.7a | 22.9 ± 2.9a | 111.2 ± 16.2a |
| | 2018 | 276 | N0 | 0 | 60.5 ± 2.5a | 15.2 ± 2.1a | 44.0 ± 9.9b |
| | | | N1 | 100 | 72.7 ± 6.7a | 23.0 ± 2.0a | 96.6 ± 9.5ab |
| | | | N2 | 150 | 73.7 ± 9.8a | 25.5 ± 4.9a | 134.4 ± 27.0a |
| *L. ovalifolium* | 2016 | 293 | N0 | 0 | 45.6 ± 4.4a | 32.5 ± 4.0a | 17.8 ± 5.8a |
| | | | N1 | 89 | 56.2 ± 2.9a | 65.2 ± 10.1a | 97.2 ± 21.6a |
| | | | N2 | 179 | 53.4 ± 7.2a | 58.7 ± 20.6a | 90.5 ± 47.9a |
| | 2017 | 279 | N0 | 0 | 24.9 ± 0.7a | 48.2 ± 11.9a | 85.1 ± 18.0a |
| | | | N1 | 33 | 19.9 ± 2.4a | 52.0 ± 7.6a | 123.5 ± 18.7a |
| | | | N2 | 67 | 22.4 ± 2.0a | 55.2 ± 13.9a | 136.5 ± 37.0a |
| | 2018 | 276 | N0 | 0 | 65.3 ± 3.9a | 121.0 ± 16a | 164.2 ± 31.0a |
| | | | N1 | 75 | 71.7 ± 4.9a | 146.3 ± 11.2a | 271.6 ± 10.6a |
| | | | N2 | 113 | 77.6 ± 5.6a | 124.1 ± 4.6a | 252.4 ± 6.8a |
| *P. laurocerasus* | 2016 | 291 | N0 | 0 | 41.1 ± 3.5a | 48.1 ± 7.5a | 26.6 ± 6.9a |
| | | | N1 | 89 | 49.0 ± 3.8a | 50.4 ± 1.9a | 28.9 ± 4.4a |
| | | | N2 | 179 | 43.9 ± 2.9a | 58.6 ± 2.6a | 37.4 ± 3.0a |
| | 2017 | 289 | N0 | 0 | 38.9 ± 1.7a | 114.8 ± 9.6a | 109.4 ± 8.1b |
| | | | N1 | 33 | 37.9 ± 1.3a | 136.0 ± 13.3a | 146.3 ± 16.1ab |
| | | | N2 | 67 | 42.6 ± 5.4a | 183.7 ± 26.1a | 193.4 ± 21.7a |
| | 2018 | 302 | N0 | 0 | 29.0 ± 2.9a | 106.1 ± 1.9a | 113.6 ± 20.3a |
| | | | N1 | 75 | 33.8 ± 1.1a | 121.4 ± 19.4a | 137.6 ± 32.1a |
| | | | N2 | 113 | 32.8 ± 2.1a | 134.8 ± 3.6a | 175.4 ± 17.2a |
| *T. cordata* | 2016 | 281 | N0 | 0 | 81.6 ± 5.6ab | 75.0 ± 5.8a | 19.7 ± 1.7a |
| | | | N1 | 89 | 91.6 ± 7.9a | 101.7 ± 17.8a | 27.4 ± 4.2a |
| | | | N2 | 179 | 60.0 ± 5.0b | 69.4 ± 19.8a | 18.7 ± 5.3a |

**Table 3.** *Cont.*

| Species | Year | DOY | N Treatment | Applied N (kg ha$^{-1}$) | Height (cm) | Aboveground Biomass (g plant$^{-1}$) | Aboveground N Uptake (kg ha$^{-1}$) |
|---|---|---|---|---|---|---|---|
| | 2017 | 277 | N0 | 0 | 179.2 ± 15.7a | 171.2 ± 43.9a | 13.0 ± 9.6a |
| | | | N1 | 33 | 214.3 ± 16.3a | 339.0 ± 83.3a | 46.9 ± 15.6a |
| | | | N2 | 67 | 172.2 ± 8.2a | 175.9 ± 14.3a | 14.1 ± 1.2a |
| | 2018 | 276 | N0 | 0 | 266.9 ± 15.5b | 437.4 ± 2.1a | 53.4 ± 1.9a |
| | | | N1 | 33 | 328.8 ± 4.8a | 497.6 ± 76.7a | 58.6 ± 13.9a |
| | | | N2 | 50 | 299.7 ± 17.3ab | 422.1 ± 63.1a | 51.1 ± 8.5a |

### 3.2. Seasonal Changes in LMA, Mass- and Area-Based Leaf N Content and Optical Leaf Measurements

#### 3.2.1. A. pseudoplatanus

Leaf mass-based N content ($N_M$) of *A. pseudoplatanus* varied between 1.96% and 3.42% (Figure 2). The smallest differences between fertilization treatments were found in 2016 and, especially, in 2018, when no differences were present only in the very beginning (DOY 123). The distinction between N1 and N2 (high N rate treatment) could only be made at DOY 163 in 2018. In 2017 and 2018, the LMA was calculated as well (Supplementary Figure S4). In 2017, the LMA increased from 60 g to around 85 g m$^{-2}$ between the two first sampling moments and fluctuated around 80 g m$^{-2}$ afterwards. In 2018, however, the LMA increased almost continuously from 40 to 90 g m$^{-2}$. Since the LMA is used to calculate area-based N content ($N_A$), the pattern also manifested itself in $N_A$ (Figure 2). Fertilization treatments resulted in similar differences between $N_A$ and $N_M$ but were more pronounce for $N_M$. The limited differences in foliar N content in 2016 resulted in hardly any difference in SPAD (Figure 3). In 2017, SPAD and Chl measured with Dualex were able to distinguish respectively, once and twice between the different fertilization levels. In 2018, a difference in fertilization could be detected with Chl measurements only for the last three sampling times. Differences in EPhen did not result in more differences in the NBI. The N0 treatment resulted consistently—with the exception of the end of the first growing season—in the lowest $N_M$, $N_A$, SPAD and Chl values and the highest EPhen values compared with the other treatments.

#### 3.2.2. L. ovalifolium

Before DOY 223, the $N_M$ of the N0 treatment of *L. ovalifolium* was significantly smaller than that of N1 and N2 (Figure 2). Afterwards, all treatments resulted in an $N_M$ of approximately 3%. *L. ovalifolium* leaves contained a similar amount of $N_M$ in 2017, with only significant differences between N0 and the other fertilizer levels at the next-to-last sampling moment. The LMA behavior was similar both years: No big differences between fertilization levels but a clear appearance of two peaks in the LMA during each growing season. Fewer differences between treatments were apparent in $N_A$ compared to $N_M$. Most differences in $N_M$ between treatments in 2016 were also apparent in SPAD (Figure 3). The only noticeable difference in $N_M$ in 2017 at DOY 265 was only detected with EPhen. In 2018, all three measuring units of Dualex (Chl, EPhen and NBI) were mostly able to distinguish between the smallest and the other two fertilization treatments when differences were present.

#### 3.2.3. P. laurocerasus

*P. laurocerasus'* leaves contained the least $N_M$ of all species; $N_M$ fluctuated around 2% (Figure 2). At the start of the first and last experimental year, the N0 treatment had a significantly lower $N_M$ than other treatments. *P. laurocerasus* showed the highest LMA of all species. In 2017, the LMA was characterized by two distinct peaks as well, in line with *L. ovalifolium* that year (Supplementary Figure S4). The LMA was generally smaller but showed a lot of in-season variability in 2018. Especially in 2017, it was clear that the course of SPAD and Chl was analogous to that of $N_A$ (Figure 3). Additionally in 2018, it could be noticed that the peaks in Chl occurred at the same moment as those of $N_A$ rather

than $N_M$. The NBI that year was able to distinguish the different N fertilization levels for all but one measuring point.

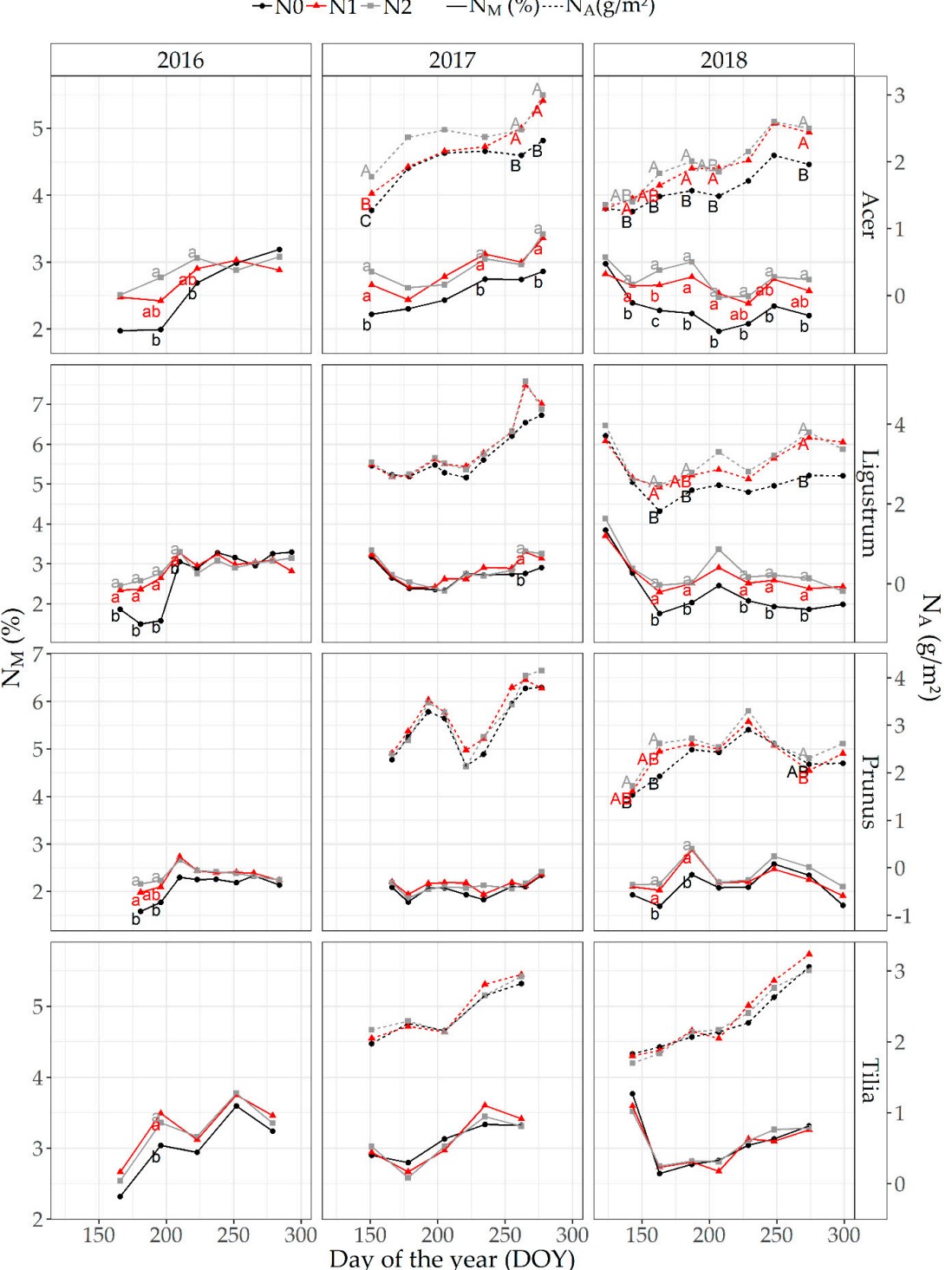

**Figure 2.** Changes in mass and area based N content of young leaves of *A. pseudoplatanus*, *L. ovalifolium*, *P. laurocerasus* and *T. cordata* during the three experimental years (2016–2018). N0: Deficiency treatment; N1: Standard treatment; N2: High N rate treatment; $N_M$ (%): Mass-based foliar N content; $N_A$ (g/m²): Area-based foliar N content. Means labelled by different letters did significantly differ at $p \leq 0.05$. If no letters occur, no significant differences were observed. Each point is the mean of three replicates of 10 leaves in 2016 and 2017 and of 15 leaves in 2018.

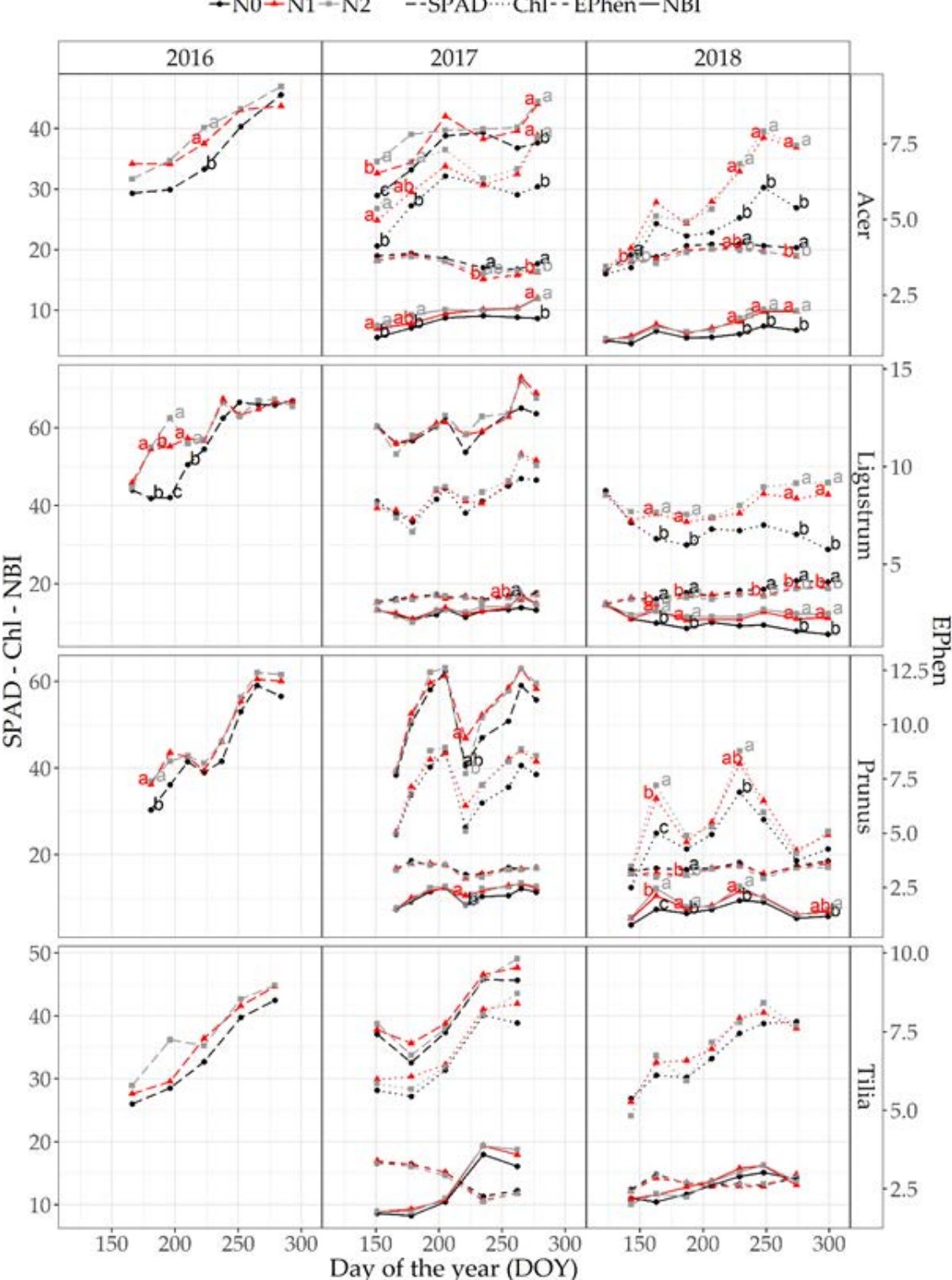

**Figure 3.** Changes in optical leaf measurements of young leaves of *A. pseudoplatanus*, *L. ovalifolium*, *P. laurocerasus* and *T. cordata* during the three experimental years (2016–2018). N0: Deficiency treatment; N1: Standard treatment; N2: High N rate treatment; Nm (%): Mass-based foliar N content; Na (g/m$^2$): Area-based foliar N content. Means labelled by different letters did significantly differ at *p* ≤ 0.05. If no letters occur, no significant differences were observed. Each point is the mean of three replicates of 10 leaves in 2016 and 2017 and of 15 leaves in 2018.

### 3.2.4. *T. cordata*

$N_M$ of *T. cordata* varied between 2.32% and 3.77% (Figure 2). In contrast with the other species, the N0 treatment only resulted in a lower $N_M$ in 2016, the year of planting. Analogously, small (insignificant) differences in leaf sensor measurements were only present in 2016. The LMA was in the same range as that of *A. pseudoplatanus* and showed a similar pattern over time.

### 3.3. Seasonal Changes in Plant Biomass, Plant N Content and Optical Plant Measurements (NDVI)

Plant N content fluctuated highly within each growing season (Figure 4). For *A. pseudoplatanus*, the mean plant N varied between 0.99% and 1.94% in 2016, between 1.33% and 2.14% in 2017, and between 1.03% and 2.13% in 2018. The plant N of N0 was lower compared with the other treatments at almost every sampling moment. It was only difficult to distinguish between the different treatments for *L. ovalifolium* 2016 and 2017. In 2016, the plant N of *L. ovalifolium* varied between 0.78% at the start (before planting, data point not in graph) and 1.91%. In 2017, the minimum and maximum values were, respectively, 1.09%—at the start—and 1.80%. Most fluctuation occurred in 2018 (0.80%–2.21%). A distinctive peak of plant N was visible for the three treatments around DOY 130. *P. laurocerasus* plants varied between 1.28% at the start and 1.93% in 2016, between 1.40% and 1.92% in 2017, and between 1.11% and 1.94% in 2018. The plant N of *T. cordata* decreased gradually with age from a maximum of 2.68% in 2016 to 1.45% in 2018 (data not shown).

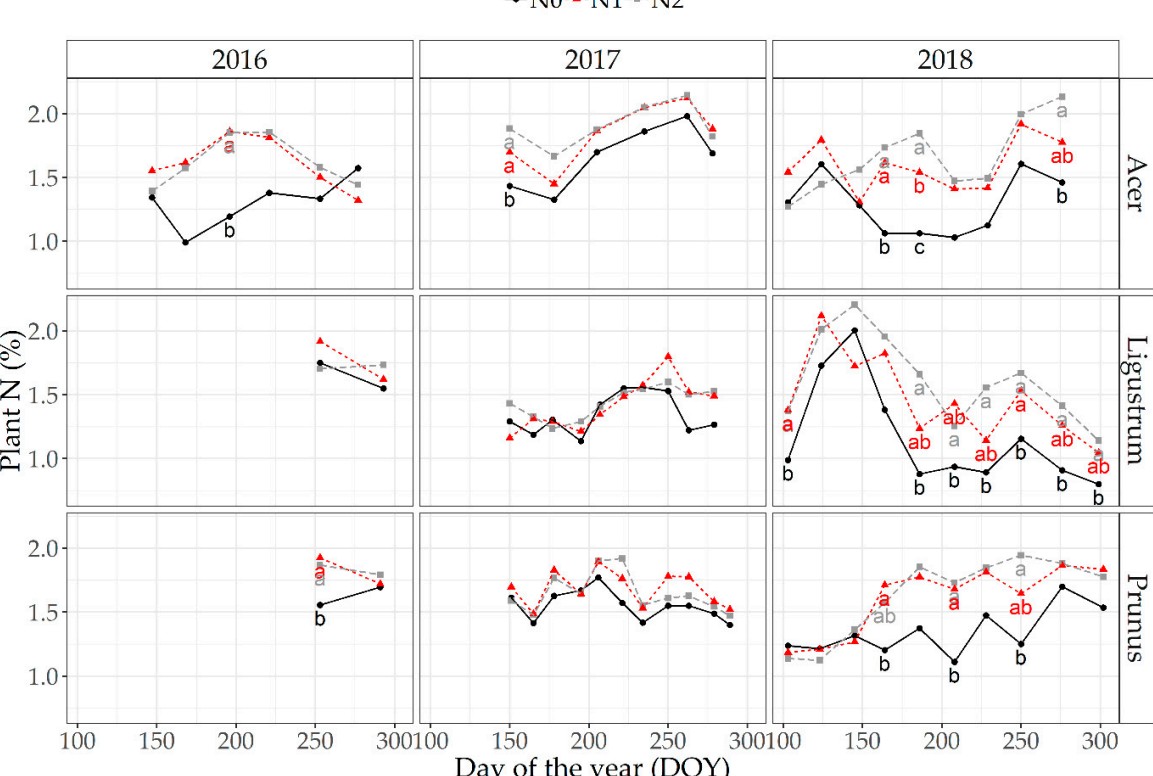

**Figure 4.** Seasonal changes in aerial plant N content (%) of *A. pseudoplatanus*, *L. ovalifolium* and *P. laurocerasus* in 2016–2018. Each data point is the mean of three replicates of pooled plants. N0: Deficiency treatment; N1: Standard treatment; N2: High N rate treatment. Means labelled by different letters did significantly differ at $p \leq 0.05$. If no letters occur, no significant differences were observed.

Figure 5 displays the seasonal changes in the NDVI of *A. pseudoplatanus*, *L. ovalifolium* and *P. laurocerasus*. The NDVI of *A. pseudoplatanus* was already ± 0.80 when measured for the first time in 2016 (DOY 168). This was not the case in 2017, when the NDVI increased gradually during the

season (DOY 178–278). In 2018, the NDVI increased between the first and the second measuring moment (DOY 124 and 148) and stagnated afterwards. In 2016, *L. ovalifolium* had an NDVI of ± 0.55 at the start (DOY 168). At DOY 194, the NDVI suddenly decreased. Two weeks later (DOY 210) the NDVI of N0 was still low while the NDVI of N1 and N2 already increased again. The minimum in the NDVI occurred at the same time as the maximum in growth increase (Supplementary Figure S3). The N0 treatment had a lower NDVI compared with the other treatments on most measuring days. Additionally at DOY 264, a small NDVI dip was present simultaneously with the second smaller maximum in growth that year. In 2017, the NDVI increased gradually from ± 0.45 to almost 0.90, with the exception of a distinct peak at DOY 222. This maximum NDVI did not coincide with a particular event in growth and might represent a measuring error that day, especially since this peak was also present for *P. laurocerasus* at the same day. In 2018, the NDVI of *L. ovalifolium* reached a plateau at ± 0.90 after the third measuring moment (DOY 164). In 2016 and 2017, the NDVI of *P. laurocerasus* could not separate the different fertilization levels. The NDVI was gradually increasing in 2016, with the exception of a minimum at DOY 202 and 210. This moment coincides with the first growth episode of *P. laurocerasus* (Supplementary Figure S3). This was not as clear for the second growth episode that year. Additionally in 2017, the growth flush around DOY 206 did not generate an abrupt change in the NDVI. A technical problem with the GreenSeeker prevented measuring around DOY 185, when the biggest growth flush of *P. laurocerasus* occurred. In 2018, the NDVI of N0 was lower compared with the other treatments between DOY 164 and 250.

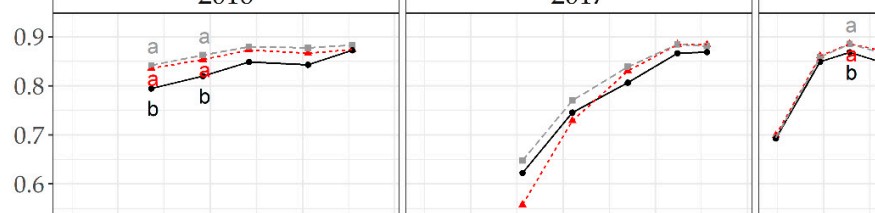

**Figure 5.** Seasonal changes in the NDVI of *A. pseudoplatanus*, *L. ovalifolium* and *P. laurocerasus* in 2016–2018. Full, dotted and dashed lines represent N0, N1, and N2 treatments, respectively. N0: Deficiency treatment; N1: Standard treatment; N2: High N rate treatment. Means labelled by different letters did significantly differ at *p* ≤ 0.05. If no letters occur, no significant differences were observed.

*3.4. Absolute/Correlation Approach*

3.4.1. Optical Measurements at Leaf Level

All measuring days were considered to select the best parameters for estimating foliar N content. Only SPAD was measured, and no LMA was determined in 2016. Good correlations ($p \leq 0.001$) were found between SPAD and $N_M$ for *A. pseudoplatanus* (Pearson's $r = 0.84$), *L. ovalifolium* ($r = 0.77$) and *T. cordata* ($r = 0.74$), but they were low for *P. laurocerasus* ($r = 0.29$, $p \leq 0.05$). The overall correlation for the four species together was low ($r = 0.24$, $p \leq 0.001$). The correlations between optically measured parameters (SPAD, EPhen, NBI) and foliar N (mass- and area-based) for 2017 and 2018 are given in Table 4. Correlations between SPAD and $N_M$ were smaller in 2017 compared with 2016, except for *T. cordata*. Considering all species together did not result in a significant correlation in 2017. The use of mass-based SPAD (SPAD/LMA) in 2017 did not improve the correlation, species-wise; however, the overall correlation became substantially stronger. Both individual correlations per species and overall correlations improved when SPAD was correlated with the area-based N concentration. SPAD and Chl measured with Dualex were highly correlated, resulting in an r with the same order of magnificence for Chl vs. $N_M$ in 2017. The overall correlation, however, was better but still low. Dualex measurements were performed on both leaf sides, but adaxial and abaxial Chl correlated similarly with $N_M$ in both years. Mass-based Chl (Chl/LMA) correlated better with $N_M$ than Chl and improved the overall correlation in 2017, in contrast with mass-based SPAD. Additionally in 2018, r increased for all species separately and combined when mass-based Chl was used. Even higher correlations were found between Chl and $N_A$. An overall r of 0.84 ($p \leq 0.001$) was found when all measurements of all four species in both years were considered. EPhen, the sum of abaxial and adaxial EPhen, (negatively) correlated generally, but not always, better with $N_M$ than Chl. When only one-sided measurements were considered, results were not consistently better on the species level. However, overall correlations for all species during the two years were stronger when abaxial EPhen ($EPhen_{ab}$) was used alone (Pearson's $r = -0.68$, $p \leq 0.001$). In contrast with the findings on SPAD and Chl, including the LMA did not improve the correlation between mass-based EPhen (EPhen/LMA) nor between EPhen and $N_A$. The combination of both Chl and EPhen by using the NBI also did not result in a higher Pearson's r compared with Chl vs. $N_A$.

In spite of high overall Pearson's correlation coefficients, the relationship between Chl and $N_A$ was species-dependent ($p \leq 0.001$). All regression equations had significantly different intercepts and, except for *L. ovalifolium*, a different slope each growing season. Mostly, a linear relationship was sufficient for describing the relationship, but in the case of *P. laurocerasus* and *T. cordata* in 2018, a second order quadratic model fitted the relationship better. The regression equations and coefficients of determination ($R^2$) are given in Figure 6.

When pooling species and growing-seasons together, a high overall r was found (0.84, Table 4), which resulted in a second order quadratic model with $y = -3.2 + 19x - 1.6x^2$. Despite a relatively high $R^2$ of 0.73, high scattering in the important region between 2 and 3 g m$^{-2}$ $N_A$ occurred. The measurements of *P. laurocerasus* were visually distinguishable from those of the other species. Therefore, they were not included in the overall regression equation presented in Figure 7.

3.4.2. Optical Measurements at Canopy Level

Table 5 presents the correlations of the NDVI vs. fresh biomass, plant N concentration, and total plant N uptake. Except for *A. pseudoplatanus* in 2018, good correlations were present between the NDVI and both biomass and N uptake. Overall correlations including all species were, however, not good. Even if *P. laurocerasus* is excluded, a low correlation was maintained ($r = 0.48$, $p \leq 0.001$). The NDVI and plant N concentration were generally not correlated well, but correlations with total N uptake per hectare were generally good. Only the N uptake of *A. pseudoplatanus* plants in 2018 was incongruous. All correlations were species- and growing-season-dependent ($p \leq 0.001$).

**Table 4.** Pearson correlation coefficients (*r*) of area- and mass-based optical parameters (Soil Plant Analysis Development (SPAD), chlorophyll content (Chl), epidermal polyphenols (EPhen), nitrogen balance index (NBI)) vs. destructively measured area- and mass-based leaf nitrogen (N) in *A. pseudoplatanus*, *L. ovalifolium*, *P. laurocerasus* and *T. cordata*.

| | *A. pseudoplatanus* | | | *L. ovalifolium* | | | *P. laurocerasus* | | | *T. cordata* | | | Overall | | |
|---|---|---|---|---|---|---|---|---|---|---|---|---|---|---|---|
| | 2017 | 2018 | 2 years [1] | 2017 | 2018 | 2 years [1] | 2017 | 2018 | 2 years [1] | 2017 | 2018 | 2 years [1] | 2017 | 2018 | 2 years [1] |
| SPAD vs. $N_M$ | 0.66 *** | | 0.76 *** | 0.49 *** | | 0.66 *** | 0.30 ** | | 0.12 ns | 0.80 *** | | 0.66 *** | -0.04 ns | | 0.10 * |
| SPAD/LMA vs. $N_M$ | 0.27 ns | | | 0.39 *** | | | 0.25 * | | | 0.72 *** | | | 0.75 *** | | |
| SPAD vs. $N_A$ | 0.80 *** | | | 0.87 *** | | | 0.84 *** | | | 0.83 *** | | | 0.86 *** | | 0.86 *** |
| Chl vs. $N_M$ | 0.60 *** | −0.04 ns | 0.28 ** | 0.45 *** | 0.49 *** | 0.48 *** | 0.32 ** | 0.07 ns | −0.02 ns | 0.81 *** | −0.04 ns | 0.31 *** | 0.22 *** | 0.30 *** | 0.26 *** |
| $Chl_{ad}$ vs. $N_M$ | 0.59 *** | −0.03 ns | 0.28 ** | 0.44 *** | 0.51 *** | 0.48 *** | 0.32 ** | 0.07 ns | −0.03 ns | 0.81 *** | −0.05 ns | 0.31 ** | 0.21 *** | 0.30 *** | 0.25 *** |
| $Chl_{ab}$ vs. $N_M$ | 0.61 *** | −0.04 ns | 0.28 ** | 0.45 *** | 0.47 *** | 0.47 *** | 0.31 ** | 0.07 ns | −0.02 ns | 0.80 *** | −0.04 ns | 0.32 *** | 0.24 *** | 0.30 *** | 0.26 *** |
| Chl/LMA vs. $N_M$ | 0.65 *** | 0.56 *** | 0.63 *** | 0.31 ** | 0.80 *** | 0.73 *** | 0.32 ** | 0.50 *** | 0.40 *** | 0.89 *** | 0.49 *** | 0.64 *** | 0.85 *** | 0.75 *** | 0.79 *** |
| Chl vs. $N_A$ | 0.89 *** | 0.93 *** | 0.91 *** | 0.84 *** | 0.77 *** | 0.83 *** | 0.83 *** | 0.85 *** | 0.87 *** | 0.90 *** | 0.80 *** | 0.82 *** | 0.86 *** | 0.79 *** | 0.84 *** |
| EPhen vs. $N_M$ | −0.72 *** | −0.66 *** | −0.72 *** | −0.35 *** | −0.60 *** | −0.56 *** | −0.09 ns | −0.53 *** | −0.36 *** | −0.83 *** | −0.33 ** | −0.55 *** | −0.49 *** | −0.53 *** | −0.51 *** |
| $EPhen_{ad}$ vs. $N_M$ | −0.77 *** | −0.58 *** | −0.71 *** | −0.14 ns | −0.73 *** | −0.55 *** | −0.20 ns | −0.45 *** | −0.32 *** | −0.80 *** | 0.09 ns | −0.37 *** | −0.02 ns | −0.31 *** | −0.19 *** |
| $EPhen_{ab}$ vs. $N_M$ | −0.64 *** | −0.64 *** | −0.68 *** | −0.34 ** | −0.51 *** | −0.49 *** | −0.03 ns | −0.53 *** | −0.36 *** | −0.80 *** | −0.64 *** | −0.70 *** | 0.74 *** | −0.64 *** | −0.68 *** |
| EPhen/LMA vs. $N_M$ | −0.54 *** | 0.28 * | −0.17 ns | −0.35 ** | 0.16 ns | 0.01 ns | −0.17 ns | 0.17 ns | 0.21 ** | −0.66 *** | 0.30 * | −0.11 ns | 0.28 *** | 0.13 * | 0.18 *** |
| EPhen vs. $N_A$ | −0.38 ** | 0.38 ** | −0.17 ns | 0.19 ns | −0.02 ns | −0.04 ns | −0.04 ns | 0.23 ns | 0.32 *** | −0.81 *** | 0.35 ** | −0.25 ** | 0.00 ns | −0.08 ns | −0.07 ns |
| NBI vs. $N_M$ | 0.80 *** | 0.12 ns | 0.50 *** | 0.58 *** | 0.72 *** | 0.66 *** | 0.35 ** | 0.19 ns | 0.05 ns | 0.84 *** | 0.05 ns | 0.47 *** | 0.46 *** | 0.51 *** | 0.47 *** |
| NBI vs. $N_A$ | 0.89 *** | 0.90 *** | 0.90 *** | 0.79 *** | 0.61 *** | 0.72 *** | 0.72 *** | 0.79 *** | 0.83 *** | 0.86 *** | 0.68 *** | 0.68*** | 0.64 *** | 0.64 *** | 0.67 *** |
| Chl vs. SPAD | 0.90 *** | | | 0.88 *** | | | 0.99 *** | | | 0.94 *** | | | 0.89 *** | | |

[1] Average values of two years are based on 2017 and 2018, except for SPAD vs. $N_M$, where they are based on 2016 (see 3.4.1) and 2017. *ns* = Non-significant, * = Significant at $p \leq 0.05$, ** = Significant at $p \leq 0.01$, *** = Significant at $p \leq 0.001$. Abbreviations: ab, abaxial leaf side; ad, adaxial leaf side; SPAD, chlorophyll measured with a SPAD meter; NM, mass-based foliar N content; NA, area-based nitrogen content; LMA, leaf mass per area; Chl, chlorophyll measured with Dualex; EPhen, epidermal polyphenol content measured with Dualex; NBI, nitrogen balance index.

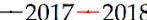

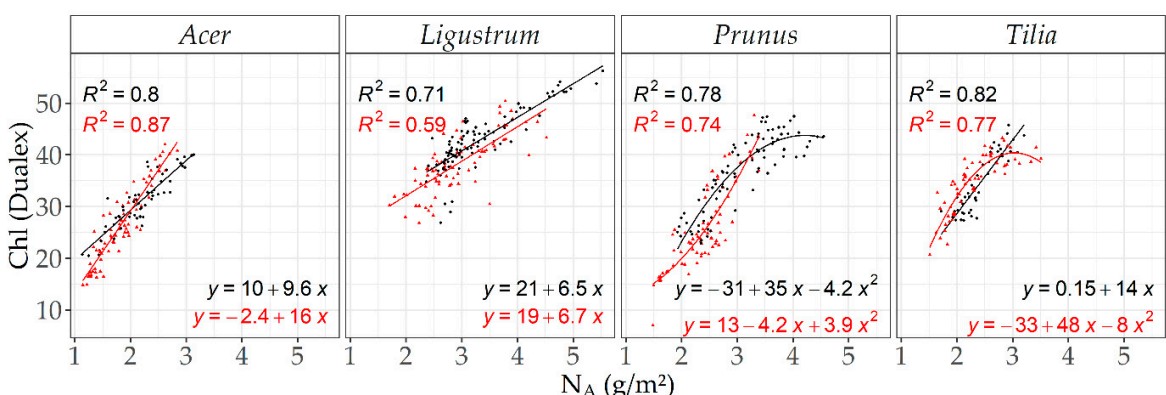

**Figure 6.** Relationship between the area-based N content ($N_A$) and Chl measured with Dualex in 2017 and 2018 for *A. pseudoplatanus*, *L. ovalifolium*, *P. laurocerasus* and *T. cordata*. Each point is the mean of three replicates of 10 pooled leaves in 2017 and of 15 pooled leaves in 2018.

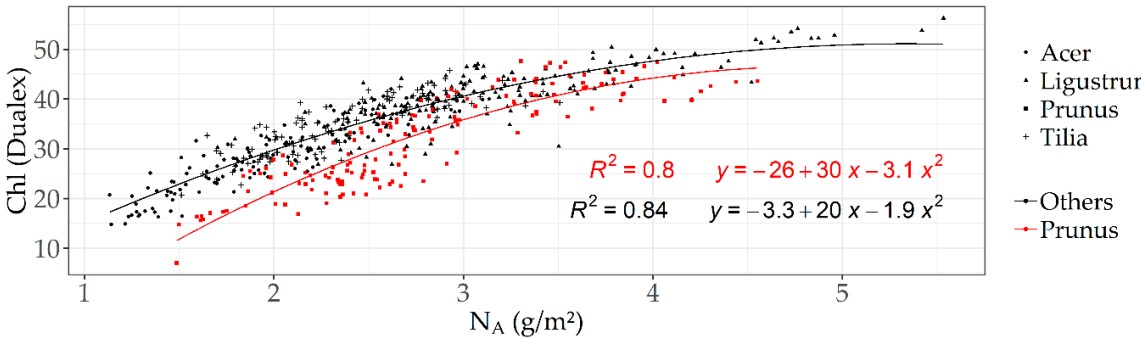

**Figure 7.** Relationship between the area-based N content ($N_A$) and Chl measured with Dualex for 2017 and 2018 combined for *A. pseudoplatanus*, *L. ovalifolium* and *T. cordata* on the one hand and *P. laurocerasus* on the other hand. Each point is the mean of three replicates of 10 pooled leaves in 2017 and of 15 pooled leaves in 2018.

### 3.5. Relative/Saturation Index Approach

3.5.1. Optical Measurements at Leaf Level

Figure 8 shows the seasonal change in the saturation index (SI) based on SPAD values in 2016. Values above 100% imply a SPAD value of N0 and/or N1 above N2. A SI of N0 was for all species below 100% on most measuring days. For all species, the SI was low early in the growing season to recover later on. Since there were not many differences between N1 and N2 in absolute SPAD values (Figure 3), the SI of N1 fluctuated around 100%. The SI of N1 dropped below 90% for L. ovalifolium and *T. cordata* on DOY 196.

Because the inclusion of the LMA improved the correlation with $N_M$, the SI based on mass-based Chl (Chl/LMA) was chosen for 2017 and 2018 (Figure 9). Similar to 2016, the SI of N1 did not differ much from 100%. Only for *L. ovalifolium* in 2018, the SI of N1 was consequently below 100%. In 2017, the SI of N0 of both *A. pseudoplatanus* and *L. ovalifolium* was under 100% at all sampling moments, which was not the case for *P. laurocerasus* and *T. cordata* that year. In 2018, the SI of N0 of all species lay below 100%. A downward trend in the SI of N0 was present for *A. pseudoplatanus* and *L. ovalifolium*, while the SI of N0 increased for *P. laurocerasus*.

**Table 5.** Pearson correlation coefficient (*r*) of the normalized difference vegetation index (NDVI) vs. destructively measured plant parameters (fresh biomass, plant N concentration and N uptake). Since only *A. pseudoplatanus* plants were harvested regularly in 2016, other species' correlations of that growing season are not shown in Table 5.

| NDVI vs. | *A. pseudoplatanus* | | | | *L. ovalifolium* | | | *P. laurocerasus* | | | Overall | | |
|---|---|---|---|---|---|---|---|---|---|---|---|---|---|
| | 2016 | 2017 | 2018 | 3 years | 2017 | 2018 | 2 years | 2017 | 2018 | 2 years | 2017 | 2018 | 3 years |
| Biomass, kg plant$^{-1}$ | 0.67 *** | 0.78 *** | 0.55 *** | 0.50 *** | 0.77 *** | 0.71 *** | 0.75 *** | 0.69 *** | 0.82 *** | 0.72 *** | 0.44 *** | 0.41 *** | 0.38 *** |
| Plant N (%) | 0.55 *** | 0.74 *** | 0.08 ns | 0.17 * | 0.47 *** | −0.44 *** | 0.03 ns | −0.19 ns | 0.66 *** | 0.28 *** | 0.31 *** | 0.08 ns | 0.15 ** |
| N uptake, kg ha$^{-1}$ | 0.72 *** | 0.79 *** | 0.49 *** | 0.53 *** | 0.75 *** | 0.59 *** | 0.69 *** | 0.69 *** | 0.81 *** | 0.74 *** | 0.69 *** | 0.59 *** | 0.61 *** |

ns = Non-significant, * = Significant at $p \leq 0.05$, ** = Significant at $p \leq 0.01$, *** = Significant at $p \leq 0.001$.

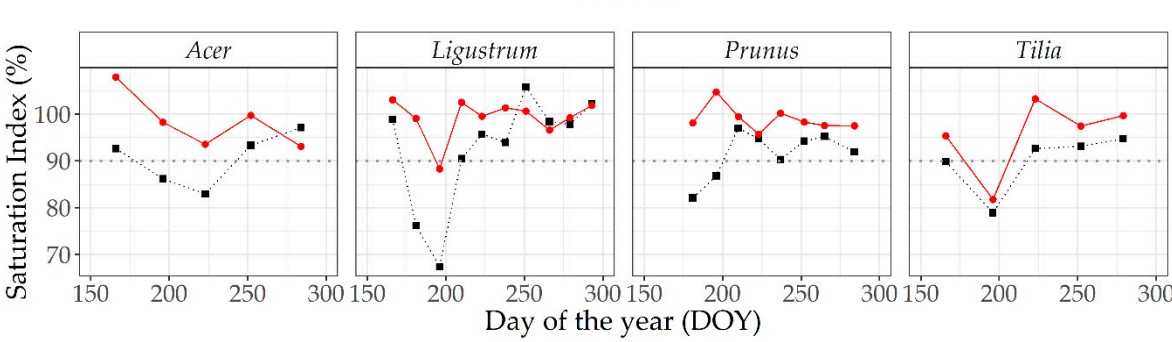

**Figure 8.** Seasonal change in the saturation index (SI) based on SPAD values of *A. pseudoplatanus*, *L. ovalifolium*, *P. laurocerasus* and *T. cordata* in 2016. N0: Deficiency treatment; N1: Standard treatment. The grey dotted line indicates an opposed action threshold.

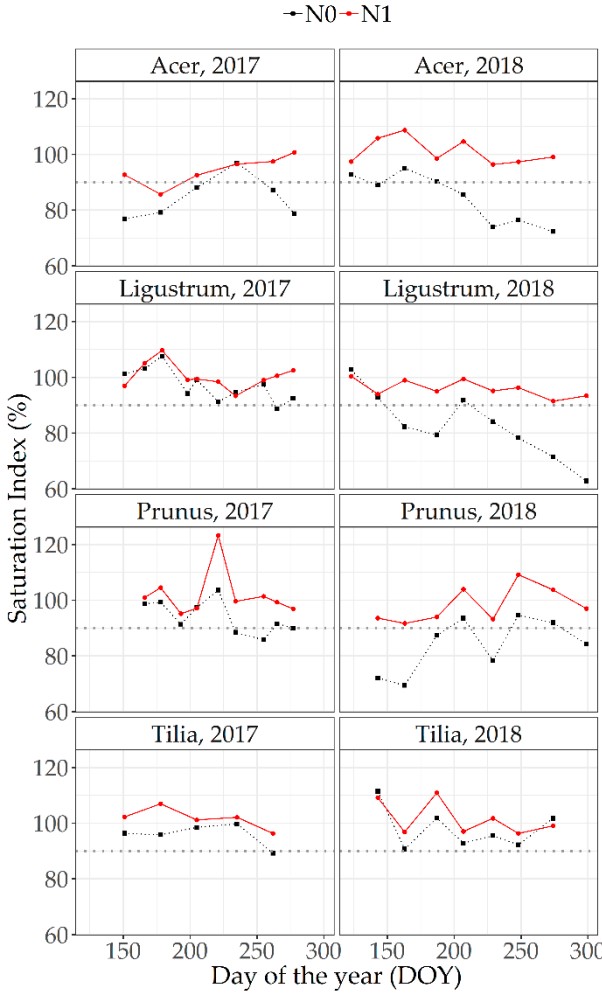

**Figure 9.** Seasonal change in saturation index based on relative mass-based chlorophyll content (Chl/LMA) of *A. pseudoplatanus*, *L. ovalifolium*, *P. laurocerasus* and *T. cordata* in 2017–2018. N0: Deficiency treatment; N1: Standard treatment. The grey dotted line indicates an opposed action threshold.

### 3.5.2. Optical Measurements at Canopy Level

Figure 10 shows the seasonal change in the saturation index based on the NDVI values in 2016–2018. Values above 100% indicate an NDVI of N0 and/or N1 above those of N2. The SI of N0 of

*A. pseudoplatanus* was below 100% at almost all measuring moments, although in 2018 the SI was close to the 100% line. The SI of the N1 of *A. pseudoplatanus* was, respectively, below and above 100% in 2016–2017 and 2018. However, differences were small except for the two first measuring moments in 2017. The SI of N0 was only at one point in time below 90%. For *L. ovalifolium*, differences were only clearly visible for N0 in 2016 when the SI of N0 dropped four times below 90% (DOY 194–236). For *P. laurocerasus*, this was twice the case in 2018 (DOY 164 and 186). For *T. cordata*, the SI was below 100% at the first two out of four measuring moments and below 90% at the second sampling moment (DOY 225, not shown).

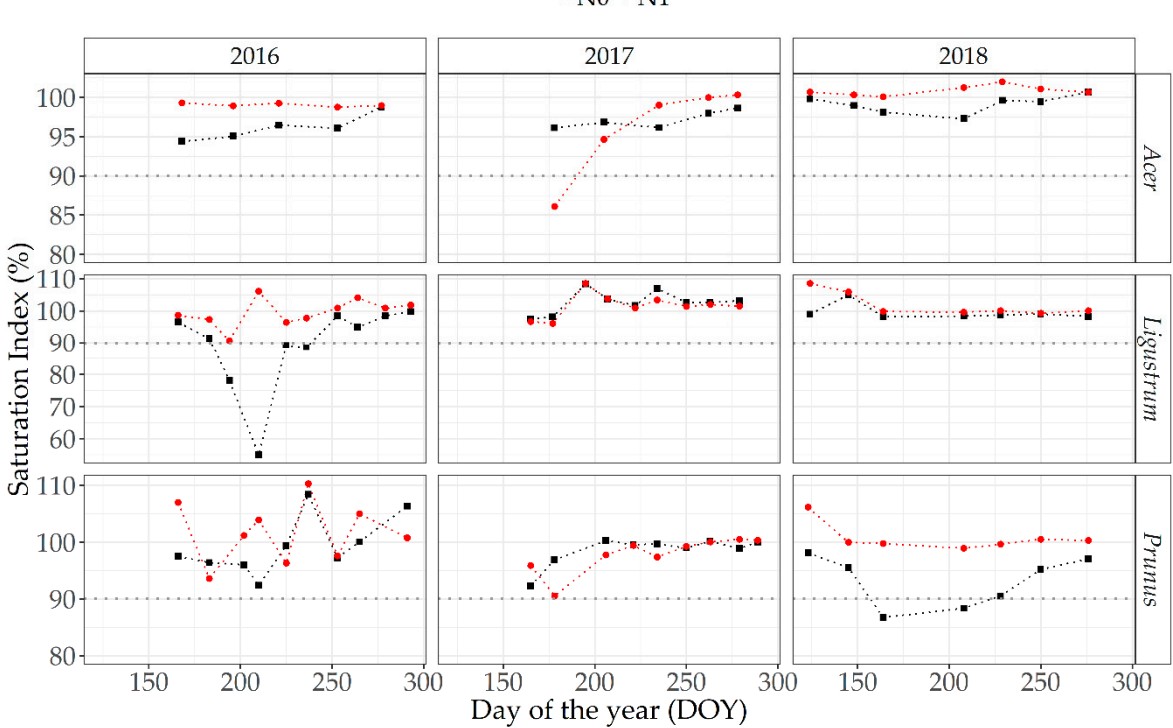

**Figure 10.** Seasonal change in the saturation index based on the NDVI of *A. pseudoplatanus*, *L. ovalifolium* and *P. laurocerasus* in 2016–2018. N0: Deficiency treatment; N1: Standard treatment. The grey dotted line indicates an opposed action threshold.

## 4. Discussion

### 4.1. Effects of Fertilization Levels

Different N fertilization treatments were applied to obtain plants with a wide range of nitrogen concentrations in order to test whether the non-destructive sensors could detect these differences. The influence of the different N levels and growing seasons on plant growth was evaluated as well. Except for *T. cordata*, the N0 fertilization treatment tended to result in the lowest biomass and the lowest nitrogen uptake every season. For *T. cordata*, a lower planting density might explain the fact that the N0 fertilization treatment did not result in a yield loss. The differences between the N2 and N1 were generally small and, in some cases (*A. pseudoplatanus* and *L. ovalifolium* 2016 and *L. ovalifolium* 2018), negative. This might indicate that the N1 treatment provided, together with natural mineralization, an ample amount of nitrogen. For *A. pseudoplatanus* in 2018, the biomass increased only slightly when the highest dose was applied, but the total uptake was well above the total uptake of the standard dose. This may indicate luxury consumption, i.e., the continued absorption of nitrogen without a further increase in plant quality [15]. Though autumn planting initially lead to a good regrowth in 2018, the dry and hot summer months negatively influenced summer growth for most plant species.

The LMA was not clearly affected by different N treatments, but it varied substantially during the growing seasons, which is in accordance with Poorter et al., who stated that the LMA is rather affected by irradiance and temperature than by nutrient limitations [37]. Since the shoot growth of *A. pseudoplatanus* fell back very early in 2018, new leaves were hardly initiated after DOY 164 (mid-June). This is reflected in the continuously increasing the LMA that year. For *L. ovalifolium* and *P. laurocerasus*, growth flushes were followed by LMA minima: Young evergreen leaves are generally lower in biomass and for that reason a decrease in the LMA took place. Only for *T. cordata*, the LMA peaks were less clearly coinciding with the occurring main shoot growth due to the length increase of axillary shoots generating new leaves as well.

## 4.2. Absolute/Correlation Approach

To evaluate the sensors on the leaf and canopy level, correlations were calculated with foliar and plant nitrogen. In 2016, the overall correlation between SPAD measurements and $N_M$ was low ($r = 0.24$) and thus not helpful to support N fertilization for a range of species. In literature [16,18,19], the LMA has been suggested to improve the correlation. Our results indeed confirm the importance of an LMA-correction, as shown in Table 4: Chl measured with Dualex was considerably better correlated with $N_A$ compared with $N_M$. This was also shown for SPAD in 2017. In cases where the LMA varied most, e.g., *A. pseudoplatanus* and *T. cordata* 2018, this is most evident. Because optical leaf sensors estimate the chlorophyll content within the sampling area of the sensor, the estimate is better linked to the LMA-corrected nitrogen content. In a denser and often thicker leaf, the same amount of chlorophyll will result in a similar value based on the area, whereas the mass-based N, i.e., N%, will be diluted and thus lower compared with a leaf with a lower LMA of the same plant species. Despite an ANCOVA analysis which pointed out a statistical difference between the two years for all species (Figure 6), Table 4 indicates that, for each species, the correlations between nitrogen and its non-destructively measured proxy became very high ($r \geq 0.82$) when the two growing seasons were considered. When the data of all four species of two growing seasons are taken into account, a good correlation was found as well ($r = 0.84$). This is an essential step for utility on a commercial nursery where various species and cultivars are grown. A quadratic polynomial model fitted for *A. pseudoplatanus*, *L. ovalifolium* and *T. cordata* resulted in an overall $R^2$ of 0.80, while the *P. laurocerasus* correlation curve indicated consistently lower Chl values for a similar $N_A$ (Figure 7). A possible explanation for this distinction is the presence of an epicuticular wax layer. This thin film reflects UV radiation, forms a mechanic and chemical barrier against insects and fungi, and reduces water loss by transpiration [38,39]. No information was vacant to compare wax thickness of *P. laurocerasus* with that of the other species. Though the *L. ovalifolium* leaves also have a glossy appearance, *P. laurocerasus* leaves are clearly thicker and more leather-like when compared with the others. That *P. laurocerasus* is an evergreen species could also influence these correlations due to a larger mesophyll volume per unit leaf area [40].

As for the relationship between epidermal polyphenols (EPhen) and $N_M$, it is known that EPhen content is negatively related to leaf $N_M$ content since their synthesis is inversely correlated [41]. This was confirmed in the present study, but the negative relationship was not as strong for all species. The best results in both years were found for *A. pseudoplatanus*. The overall correlations between EPhen and $N_M$ were generally better compared with correlations between Chl and $N_M$. LMA corrections did not improve the correlation between EPhen and $N_M$; epidermal volume was less affected by a changing LMA than mesophilic compounds such as chlorophyll [40]. The main advantage of EPhen vs. $N_M$ over Chl vs. $N_A$ is that the LMA does not have to be determined separately, making the method more convenient for in situ use. The NBI and $N_M$ were not better correlated than Chl and $N_A$ or EPhen and $N_M$, especially in 2018 when only a significant correlation was found for *L. ovalifolium*. Additionally, the correlation between the NBI and $N_A$ was less strong than between Chl and $N_A$ alone, proving that the NBI does not add an additional value over Chl alone.

Though we showed that sensor values correlated well with leaf N for all species, it is difficult to link sensor output to management decisions regarding N side-dressing, especially since absolute thresholds are species- and cultivar-specific and, moreover, variable over time. For *Acer pseudoplatanus* and *Tilia cordata*, respectively, 2.2% and 2.0% were reported as critical values for $N_M$ in urban green [42]. No sufficiency values were found for *Ligustrum ovalifolium* and *Prunus laurocerasus*. For EPhen, no optimal values are described in literature as well. In 2016, the $N_M$ of N0 of *A. pseudoplatanus* was below 2.2% on the first two measuring moments. It is therefore likely that additional N uptake could have prevented the lower biomass of N0. Foliar N deficiency appeared before biomass accumulation was affected, demonstrating the value of foliar analysis as decision support tool. Though $N_M$ of *A. pseudoplatanus* was never below 2.2% in 2017, there was a small difference between the fertilization levels. $N_M$ of N0 was in 2018 clearly deficient from the second to forth measuring moment, but only below 2.2% at the two later sampling moments. Thus, for *A. pseudoplatanus*, our results suggest a higher threshold value of 2.5%. When assuming 2.0% as a minimum value for *L. ovalifolium* and *P. laurocerasus*, as our results suggest, yield loss could possibly been avoided when reacting in 2016 and 2018. In 2017, *P. laurocerasus* leaves of all three fertilization levels were continuously fluctuating around 2.0%. An additional side-dressing, on top of the 67 kg N ha$^{-1}$ already supplied, would probably have resulted in an excessive amount of $N_M$ for N2. The foliar N of *T. cordata* was below 2.0% in none of the three years, and no differences between the fertilization levels could be reported after three years. The low planting density, typical for the cultivation of avenue trees, ensured an adequate N supply by natural mineralization, in addition to allocation from perennial storage organs after winter. Rather than working with sufficiency values related with $N_M$, absolute sufficiency values determined by using yield response are described in literature [43,44]. However, these approaches are species specific and therefore less convenient for nurseries.

In this study, the utility of both optical sensors on the leaf and canopy levels was examined. While leaf sensors are promising in predicting leaf N content, plant N content was not estimated accurately by the NDVI. Plant biomass and total N uptake per hectare were, on the other hand nicely, correlated with the measured the NDVI. *A. pseudoplatanus* in 2018 suffered most from the challenging weather conditions in May–July when natural rainfall was exceptionally low (Table S3), resulting in low correlations that year. In Figure 5, one could already see that the NDVI tends to saturate over time as the canopy closes. To predict plant N% accurately over time, the NDVI was affected too much by the expanding green biomass. For plants with a narrow planting distance (e.g., *A. pseudoplatanus*) or a high start growth (e.g., *L. ovalifolium* in 2018), this occurred early in the growing season. On the other hand, sensor measurements of widely spaced plants are possibly influenced by the background reflection of bare soil. As such, only a relative approach can be of use. Because growth measurements were carried out for all species, we were able to link some changes in the NDVI to new growth. Young and not fully emerged leaves were different in color compared to older leaves. Especially for woody plants with an episodic growth, sudden growth flushes might generate a large amount of new leaves, affecting the correlations between the NDVI and other parameters of interest. Considering the good correlation with biomass and N uptake, the NDVI can be used to assist growers in making site-specific decisions. When an entire field is scanned, the NDVI can identify locations in the field that need additional attention. Whether this lower NDVI is due to an N limitation in the soil is, however, uncertain.

### 4.3. Relative/Saturation Index Approach

Besides an absolute approach, where the sensor output values were compared with threshold values, a relative approach was considered. To this end, a saturation index was calculated wherein N2 was used as a so-called reference plot with ample fertilization. When the SI deviates severely from 100%, adjustments should be made. This approach is less susceptible to in-season variation and has the advantage that, when all other parameters but N are constant, differences in the SI will only indicate an N deficit. To avoid excessive fertilization, it is still important to choose the fertilization dose of the reference plot carefully. When luxury consumption occurs, the saturation index approach will suggest

a false deficiency. Since our results points to inconsistencies in regression equations over different seasons and species, this approach might be more convenient for nurseries. It is nevertheless still necessary to point a threshold value for action. The action threshold of 90% that we used was opposed by Varvel et al. [25]. In 2016, both the SI and absolute values for SPAD of N0 *A. pseudoplatanus* would suggest side-dressing. On the contrary, in 2017, $N_M$ was never below 2.2%. The relative assessment would, in this case, have prevented yield loss, since, at the first three measuring moments, the SI was under 90%. The SI in 2018 indicated a deficit earlier compared with the absolute approach. Analogously as for *A. pseudoplatanus*, both approaches resulted in roughly the same results for *L. ovalifolium* and *P. laurocerasus* in 2016. For *L. ovalifolium* in 2017, neither method was satisfactory, but differences between fertilization levels were small that year. In 2018 however, the SI approach indicated deficits early in time and before the absolute values would have. For *P. laurocerasus*, the SI approach might have prevented yield loss in 2017 and would definitely have prevented yield loss in 2018.

Since the implications of a zero fertilizing strategy on the yield of *A. pseudoplatanus* are apparent, the 90% threshold is clearly not sufficient on the canopy level. In 2016, the SI based on the NDVI would have been only helpful for *L. ovalifolium*. Despite the fact that the SI on the leaf level was appropriate for both *L. ovalifolium* and *P. laurocerasus* in 2018, the method on canopy level only suffice for *P. laurocerasus*, although 90% here seems slightly too high. This is due to the early saturation of the NDVI when the canopy closed.

## 5. Conclusions

Three years of intensive measurements showed that proximal optical sensors can successfully estimate leaf N content in the studied woody ornamental plants. However, correlations were only sufficient and consistent over the different growing seasons when the LMA was implemented. Best correlations were found for all species between Chl and SPAD vs. $N_A$. This makes it possible to use proximal leaf sensors to estimate actual N content of the leaves relatively fast compared to lab-analysis. Nevertheless, care should be taken when different species are used, especially when leaf and/or plant characteristics differ, as our results pointed out for *P. laurocerasus*. On the canopy level, the NDVI determined with the Greenseeker did not reliably predict N content, but it can be of help to correct for in-field variability and support on-farm decisions. To interpret sensor measurements and determine the utility of decision support systems, two approaches were tested. At the nursery level, a high assortment of woody ornamentals was produced, and a relative approach was found to be more useful compared to an absolute approach. The relative approach at the leaf level can prevent quality loss in most cases, mainly because the sufficiency values for absolute use are scarce. As a result of sensor output saturation in case of early canopy closing and integration of background signals for widely spaced crops, both approaches are generally of less use at the canopy level in nurseries. To conclude, proximal optical sensors show potential to improve the fertilization of woody ornamentals and can help to control residual N levels in open field production of hardy nursery stock.

**Supplementary Materials:** The following are available online at http://www.mdpi.com/2073-4395/9/7/408/s1, Figure S1. Dimensions of one experimental plot in 2016 (left) and in 2017 and 2018 (right). Table S1. Corresponding dates for occurring Days Of the Year (DOY) on graphs. Table S2. Fertilization dates in 2016–2018. Table S3. Average precipitation (mm) and air temperature (°C) during the three growing seasons at the experimental site and 30-year average data from RMI Belgium. Ukkel (1981–2010). Figure S2. Biomass accumulation for *A. pseudoplatanus*, *L. ovalifolium* and *P. laurocerasus* at each experimental year (2016–2018). Figure S3. Length increase for *A. pseudoplatanus*, *L. ovalifolium*, *P. laurocerasus* and *T. cordata* at each experimental year (2016–2018). Figure S4. Changes in LMA of young leaves of *A. pseudoplatanus*, *L. ovalifolium*, *P. laurocerasus* and *T. cordata* in 2017–2018.

**Author Contributions:** Conceptualization, J.B. and M.-C.V.L.; methodology, J.B., S.A., and M.-C.V.L.; data collection, J.B., S.A., and L.S.; analysis: J.B.; writing—original draft, J.B.; writing—review and editing, M.-C.V.L., H.V., A.E., L.S., and S.A.

**Funding:** This work was supported by VLAIO (Flanders Innovation and Entrepreneurship; project 1406966).

**Acknowledgments:** We wish to thank all the technical staff of PCS Ornamental Plant Research for their technical support.

**Conflicts of Interest:** The authors declare no conflict of interest.

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
