# Peer review of "Application of Proximal Optical Sensors to Fine-Tune Nitrogen Fertilization: Opportunities for Woody Ornamentals"

_agronomy, doi:10.3390/agronomy9070408_

Round 1

Reviewer 1 Report

GENERAL CONSIDERATIONS:

The paper covers a very interesting topic. Data are many, and this makes it difficult sometimes, to follow the result presentation even if they are correctly described. Of course, that was even harder because of the “Error! Reference source 263 not found” troubles all along the paper.

Other minor problems were found in the materials and methods section in particular regarding:

- how the amount of the fertilizers were determined

- when the fertilizers were applied

- missing information about the irrigation

- some others.

(In the pdf manuscript file you will find some point by point comment/correction/request of information).

However, the main point in my opinion is the number of plants sampled to evaluate biomass accumulation. For all species in 2016 and for two species in 2017 only one plant per plot was sampled. This means that the presented values derive from the average of only three plants per treatment. In my experience that is too little to make supposition on the response of the species to the applied treatments. This might be the reason why so little differences in table 3 were found even if for almost all species and years there was the tendency for improving performances along with the increase of applied N.

Otherwise I do not see any reason why I should fertilize these plants (as you would suggest using the relative saturation index approach) if there are no plant growth and quality improvement.

Because of this, all the speculation linked to the effect of fertilization treatment on biomass production should be “softened” or even omit some data on plant biomass (e.g. aboveground biomass accumulation figure S2)

Reviewer 2 Report

The manuscript ‘Application of proximal optical sensors to fine-tune nitrogen fertilization: opportunities for woody ornamentals’ describes the feasibility of fluorescence and hyperspectral measurements as non-invasive measure to determine plant nitrogen status. Based on correlations between optical determinations and leaf harvest properties, they calculated an ‘simulation’ index to support an accelerated estimation of plant nutritional status and thus for optimizing nitrogen fertilization of four woody species. The analytical evaluation of three extensive seasonal experiments and the calculated ‘simulation index’ might help to improve urban and horticultural cultivation of woody species. The experiment has followed a correct procedure, the manuscript is well written in general, and statistical analysis has been performed appropriately.

I have some concerns, which need to be addressed before considering for final publication. In general, the authors’ goals could be stated more clearly in the discussion and conclusion part. Additionally, as a fundamental point the errors in citing need to be corrected (references not correctly linked), while missing line numbers (starting from page 16) made it difficult to refer to the appropriate text parts.

With major changes this manuscript will fit well into the scope of agronomy and offers new insight into tools for a nitrogen-saving ornamental production as well as into data analysis for other agricultural areas.

In more detail:

-          Line 91 ff: Here for me, the main message is not quite clear. What exactly is meant with “non-limiting area” and how does it stand in contrast to “sensor measurements”? As mentioned by the studies [21,22], SPAD measurements were standardized by comparing a N-deficient field with a control field (optimal N) supply. So, the authors of [21,22] suggest to refer to a reference plot with ample N, with the aim to determine if the tested plot has less/higher N content. However, this is a limited area, at least for my understanding.

-          Line 137: JD is an unusual counting of dates for non-astronomers. “Day of the year” might be more appropriate, particularly when starting each year with 1 JD.  Originally, the Julian Day is a continuously ongoing counting of days starting from 4713 BC.

-          Line 159: Please add information (company etc.) to calcium ammonium nitrate, and Tropicote.

-          Line 162: Please add information (company etc.) to Patentkali missing.

-          Line 164: table1: It would be helpful to read a description of the nitrogen applications N0, N1 and N2 in the text of M&M part, not solely in “abbreviations” of table1.

-          Line 178: See above, first wrong reference “error”

-          Line 184: Authors could consider to describe the N determination by Dumas combustion analysis in a bit more detail.

-          Table2: The table caption is already very detailed, however, for me, it is not quite clear how many trees have been harvested and measured. E.g. Acer in 2018 “5 (10)”, does it mean that 5 trees were harvested and there were 10 trees at the beginning, so 5 were left over at the end of the experiment? It might be considered to add some more explanation.

-          Line 193: wrong reference source “error”

-          Line 215: Please add information about the device and the experimental setup of measurements. How exactly was the distance kept of 80-100 cm? was the device handled by hand or part of a ‘fixed’? Please add some more technical properties of this measurement device.

-          Line 232. Wrong reference “error”, in the following all citations show errors. [will not be mentioned in the following again]

-          Line 306: table 2: Please be consistent in legend and description compared with manuscript text, NM and NA (big letters) instead of Nm and Na

-          Page 16: No further line numbers given

-          Page 17:  citation problem, figure shown instead of reference text

-          Page 18: citation problem, figure shown instead of reference text

-          Page 20: At the beginning of the discussion part, I miss the authors’ hypotheses/goals/aims.

-          Page 20, 4.1. line 1-3, might it be possible that Tilia has a lower N demand, in general, and therefore N0 did not result in growth/Chl losses? Compare N uptake described e.g. in https://onlinelibrary.wiley.com/doi/epdf/10.1002/jpln.201000004, https://www.sciencedirect.com/science/article/pii/S0304423802001516.

-          Page 21, line 6: sentence too long and/or grammatically not correct.

-          Page 21, line 23. Sentence “overall…” different points arranged into one sentence.

-          Page 21, line 44: “only in 2017…” sentence grammatically not correct.

-          Page 22, 4.3. The authors calculated the saturation index by dividing N0 or N1 by N2 (higher N treatment). In general, I would expect to use the standard treatment (N1) as reference, as also shown by Varvel et al. 1997 (correct the publishing date, it is not 2010). While the authors discussed N2 as ample treatment with mentioned advantages in seasonal calculations and state that changes should be made, when SI below 100% (which is the level of more than the standard N level). A subsequent 90% threshold for action might help to overcome this issue, but might still be misleading. Therefore, the sentence “when SI deviates from 100%, adjustments should be made” is not completely true to optimize N supply, instead it might be changed into “when SI deviates from 90%” or threshold might be calculated with N1 (standard treatment).

-          How does the statement “relative approach will prevent quality loss” match with the last sentence “approaches are generally of less use on canopy level in nurseries”? I guess, this refers to the different devices, but at least for me, it is not quite clear which device might help to improve nitrogen fertilization and which would not.

-          References: The authors should check their references for correctness, at least Varvel et al. [25] was rather published in 1997 than 2010.

Round 2

Reviewer 1 Report

There are very few corrections to make to the text.

Reviewer 2 Report

The manuscript has been significantly improved. All mentioned points were targeted appropriately.